# High temperature environment reduces olive oil yield and quality

Yael Nissim[1], Maya Shloberg[1,2], Iris Biton[1], Yair Many[1], Adi Doron-Faigenboim[1], Hanita Zemach[1], Ran Hovav[1], Zohar Kerem[2], Benjamin Avidan[1], Giora Ben-Ari [1]*

1 Institute of Plant Sciences, ARO, The Volcani Center, Rishon LeZion, Israel, 2 Institute of Biochemistry, Food Science and Nutrition, The Robert H. Smith Faculty of Agriculture, Food and Environment, The Hebrew University of Jerusalem, Rehovot, Israel

* giora@agri.gov.il

## Abstract

Global warming is predicted to have a negative effect on plant growth due to the damaging effect of high temperatures. In order to address the effect of high temperature environments on olive oil yield and quality, we compared its effect on the fruit development of five olive cultivars placed in a region noted for its high summer temperatures, with trees of the same cultivars placed in a region of relatively mild summers. We found that the effects of a high temperature environment are genotype dependent and in general, high temperatures during fruit development affected three important traits: fruit weight, oil concentration and oil quality. None of the tested cultivars exhibited complete heat stress tolerance. Final dry fruit weight at harvest of the 'Barnea' cultivar was not affected by the high temperature environment, whereas the 'Koroneiki', 'Coratina', 'Souri' and 'Picholine' cultivars exhibited decreased dry fruit weight at harvest in response to higher temperatures by 0.2, 1, 0.4 and 0.2 g respectively. The pattern of final oil concentration was also cultivar dependent, 'Barnea', 'Coratina' and 'Picholine' not being affected by the high temperature environment, whereas the 'Koroneiki' and 'Souri' cultivars showed a decreased dry fruit oil concentration at harvest under the same conditions by 15 and 8% respectively. Regarding the quality of oil produced, the 'Souri' cultivar proved more tolerant to a high temperature environment than any other of the cultivars analyzed in this study. These results suggest that different olive cultivars have developed a variety of mechanisms in dealing with high temperatures. Elucidation of the mechanism of each of these responses may open the way to development of a variety of olives broadly adapted to conditions of high temperatures.

## Introduction

Fluctuations in temperature occur naturally during plant growth and reproduction. However, extreme hot summers can damage the intermolecular interactions needed for proper growth, thus impairing plant development and fruit set. The increasing threat of climate change is already having a substantial impact on agricultural production [1]. High temperatures may cause visual symptoms of sunburn, leaf abscission and growth inhibition of plants [2, 3]. High

---

**Data Availability Statement:** All data is in the supplementary tables.

**Funding:** This work was supported by a grant (No. 2031170) of the Israeli Ministry of Agriculture and Rural Development to GB.

**Competing interests:** The authors have declared that no competing interests exist.

temperature shocks during the reproductive phase of cereal crops can cause substantial reduction in yield. Oilseed crops are also negatively affected by heat stress, which has been shown to reduce starch, protein and oil content [4, 5]. Significant reduction of yield due to heat stress has been reported in peanut and tomato [6, 7] and due to high temperatures has been reported in wheat, rice and bean [8–10]. Plants exposed to temperatures above their optimal growing temperatures, exhibit cellular and metabolic responses which enable the plants to survive [11–15]. The level of damage to crops caused by high temperatures depends on the growth stage at the time of exposure, and the severity of the stress. The reproductive phase is more sensitive to high temperatures, causing a reduction in yield [16].

As in other drupes, olive fruit development is characterized by a double sigmoid growth curve for fruit size and weight [17, 18]. Rallo and Rapoport [19] examined the development of 'Manzanilla' olive fruit mesocarp. They found that as in other drupes, both cell division and expansion contribute to initial mesocarp growth. From six weeks post anthesis, mesocarp growth is determined solely by cell expansion. Hence, the early difference in fruit size between cultivars is mostly a result of the rate of cell division. Later fruit growth is a function of increased cell size [20].

Oil accumulation in olive fruits is known to begin in the second half of the summer. It lasts for about 8 weeks during summer and fall and then slows down during fruit ripening. Oil accumulation is strongly influenced by the cultivar type and the climactic conditions prevalent during fruit development. During ripening, the olive fruit undergoes various modifications [21–24].

The period of drupe development and maturation lasts 18–20 weeks after flowering (WAF). During this period, the drupe passed through four developmental stages. The first stage begins soon after fertilization, lasts for 2 weeks and is characterized by ovary growth through cell division. The second stage lasts up to 6 WAF, and is characterized by drupe growth by cell expansion. The third stage extends for 7–10 WAF. During this stage, the fruit continues to grow by cell expansion. Lipids are stored in oil bodies in the mesocarp cells. The first oil bodies (2–4 per cell, on average) appear at about seven WAF. As the fruit matures, oil bodies increase in number and begin to coalesce into one main oil droplet per cell, occupying $\sim$40% of the cell volume, and to several minor oil bodies. By the completion of pit hardening at about 10 WAF, the oil bodies have fused into a large oil droplet comprising $\sim$60% of the cell volume. The fourth stage at 10–20 WAF is characterized by continued oil accumulation and at about 16–17 WAF the color of the epicarp begin to change to purple due to intense anthocyanin deposition in the vacuole. At that time, the oil droplets occupy about 80% of the cell volume [25–28].

Virgin olive oil is characterized by its sensorial and nutritional properties, which are different from those of other plant oils. Its health benefits are due to both its fatty acid composition and minor compounds such as polyphenols, tocopherols, pigments and vitamins. Olive oil contains mainly monounsaturated fats. The main fatty acid is oleic acid (C18:1), which represents between 55 and 83% of the total oil. It also contains a moderate amount of linoleic acid (C18:2; 3.5–21%) and palmitic acid (C16:0; 7.5–20%) and a small amount of stearic acid (C18:0) and linolenic acid (C18:3). Fatty acid composition of olive oil is strongly affected by several agronomical factors such as cultivar type, fruit ripeness, crop yield, and growing medium [29]. Several studies have examined the effect of high temperatures over a long period of time on olive fruit weight [30, 31], olive oil accumulation [30–33] and olive oil fatty acid composition [30, 33–37]. Garcia-Inza et al. [30] studied the effect of temperature during the oil accumulation phase on these parameters, using transparent plastic chambers with individualized temperature control to manipulate temperatures. One experiment was carried out for four months with four treatments of different temperatures ranging from16.7 to 30.7˚C. Another experiment was carried out for four successive one month long treatment periods

with two treatments at differing temperatures. They found that fruit dry weight was not affected by average temperatures within a range of 16–25˚C, but was reduced with further increases in temperature. Oil concentration decreased linearly at $1.1\%\,{}^{\circ}C^{-1}$ across the entire range (16–32˚C) of average seasonal temperatures explored, while oleic acid concentration decreased $0.7\%\,{}^{\circ}C^{-1}$ over the same range. In the one month long experiment, an additional 7˚C above the control had a permanent negative effect on oil concentration at final harvest, particularly when the exposure to high temperature was at the beginning of oil accumulation. Oleic acid concentration was also negatively affected by the high temperature treatment. However, oleic acid concentration recovered upon removal of the chamber, with the exception being that just as with total oil accumulation, oleic acid accumulation was retarded when the heat treatment was applied at the beginning of fruit development. In general, they concluded that high temperatures during oil accumulation negatively affected olive oil yield and quality in warm regions, particularly if the high-temperature event occurs early. Garcia-Inza et al. [30] used chambers which may distort the effects of a natural environment. Other studies tested the effect of high temperatures comparing fruits of various cultivars in different locations during different years. When comparing the yields of trees from different orchards, which were subjected to different methods of agriculture, many factors can influence the parameters being tested. In order to avoid these problems, we compared adult plants in pots, all of the same age, and whose fertilization, pruning and other treatments were equal, the only difference between the two groups being their location during the period of fruit development.

Olives are a major crop in Israel and are planted all over the country including semi-desert regions such as the area surrounding Tirat Zvi, where summer temperatures often rise above 40˚C. The objective of this study was to characterize the effect of a high temperature environment during olive oil accumulation, on oil yield and quality. In order to address this we used five year old potted olive trees of five selected cultivars and placed one group in Tirat Zvi (high temperatures during the summer) during the period of fruit development, and a second group in Tzuba, a village with relatively mild summers, for two consecutive years. Characterizing fruit development, oil accumulation and oil quality in both groups revealed that unsurprisingly, high temperatures during oil accumulation can decrease fruit size, oil yield and oil quality. We demonstrated however, that sensitivity to high summer temperatures is genotype dependent.

## Materials and methods

### Experimental design

The olive cultivars 'Barnea', 'Coratina', 'Koroneiki', 'Souri' and 'Picholine' were used in this study. Twelve olive trees from each of the five cultivars brought from the Boxer nursery (Bnei Darom, Israel) and planted in pots in 2010 at the Volcani Center, Israel (31˚59'N 34˚49'E; 42 m above sea level). During the summer of 2015, the trees were replanted in 50 liter pots. Right after fruit set, at the beginning of May 2016, six saplings of each cultivar were placed in an extremely hot climate zone at Tirat Zvi village (32˚25'N 35˚31'E; 225 m below sea level). The remaining six trees of each cultivar were placed in a more temperate environment at Tzuba village (31˚47'06.0"N 35˚06'33.5"E; 705 m above sea level). Both locations are private land and the owners of the land gave permission to conduct the study on these sites. Temperature measurements were determined using a HOBO USB Micro battery-powered station, which is a weatherproof data logger for monitoring temperature. The HOBO devices were stationed in the shade at both locations and measured the temperature every two hours throughout the entire period of the experiment. Data regarding humidity, rainfall and wind speed was taken from the Israel Meteorological Service database (https://ims.data.gov.il), measured at meteorological stations located 5 km from Tirat Zvi and 4.5 km from Tzuba.

The plants were watered using a drip irrigation system which delivered 2L of water per hour spread over three 25 minute cycles per day (07.00, 12.00 and 17.00 h) at Tzuba, the moderate temperature (MT) site, and four such cycles (06.00, 10.00, 14.00 and 18.00 h) at Tirat Zvi, the high temperature (HT) site. Each plant was irrigated with four drippers. An additional fifteen minutes of irrigation was added at each location in mid-summer to ensure an adequate supply of water. Irrigation was delivered in excess, in order to prevent dry soil. The pots had perforated bottoms in order to allow the escape of excess water. Taking into account the differences in evaporation between the two locations, the trees in the HT site were watered one third more than the trees at the MT site. In total, each tree in the HT site was watered with 13.3 liters per day and the trees in the MT site with 10 liters per day per tree. During July and August 2 liters per day per tree were added in both locations. Weeds were controlled by manual removal. Olive fruit fly control was done only in Tzuba site using Vertimec EC, Vertigo EC and Score EC as necessary, as well as Rogor L-40 EC which was applied once a month throughout the experiment. Plants were fertilized at the beginning of May with Osmocote, smart-release plant food; one application contains 11 essential nutrients and is effective for 6 months.

In order to address the effect of a high temperature environment on olive fruit development as well as oil accumulation and quality, we characterized various physiological parameters of the five olive cultivars 'Barnea', 'Picholine Languedoc', 'Koroneiki', 'Souri' and 'Coratina' during the fruit development period of two consecutive years. Trees of each cultivar were placed at two different locations, two weeks after full bloom: The HT site (Tirat Zvi), was characterized by very warm summers and the MT site, Tzuba, with a mild summer. Temperatures were measured at both locations every two hours during the entire season by portable data loggers. Olives from each of the five cultivars on both locations were sampled every month throughout the experiment for physiological and histological analysis as well as for evaluation of olive oil content. At the end of the season, all fruits remaining on the trees were harvested, the oil extracted and analyzed for quality. Trees were returned to the Volcani center for the winter and spring, and in May 2017, two weeks after full bloom, the trees were transported once more to Tirat Zvi (HT site) and Tzuba (MT site). Transportation in both years was carried out very carefully and no loss of fruit was experienced as a result of the move. In total, the experiment lasted from the beginning of May till the end of October 2016 and from the beginning of May till Mid December 2017. At the end of the 2016 season, we harvested only a limited number of 'Souri' and 'Picholine' fruits. Therefore, the 'Souri' fruits were only partially analyzed and the 'Picholine' fruits were not analyzed at all. In 2017, all five cultivars were fully analyzed. Fruit load was not measured, even though it may have had an effect on fruit weight and oil concentration. However, plants were randomly chosen to be placed in either of the two locations and therefore we believe that the average fruit load was similar at the HT and the MT sites.

## Evaluating olive oil concentration

Whole fruit fresh weight was recorded, and the fruit was then dissected to mesocarp and seed for further analysis. The fresh weight of each dissected section was recorded. Olive mesocarp and endocarp was oven dried at 90˚C for 48 h and the dry weight was recorded. Oil content (dry weight basis) was determined using chemical oil extraction with petroleum ether as a solvent in quintuple (approximately 5 gr each).

## Cell size—histological analysis

A section of the fruit was analyzed for the number of cell layers (S1 Fig) and cell area using differential staining. Fresh fruit sections from each sampling date were preserved throughout the experiment using FAA (Formaldehyde 10%, Ethanol 50%, Acetic Acid 5% and water 35%) as a

fixative. At the end of the experiment, we analyzed only samples representing different developmental stages. The beginning of June, 50 days post anthesis (DPA), represented the developmental stage preceding pit hardening. The beginning of July, 83 DPA, represented the developmental stage just after pit hardening, and September, 146 DPA, represented the period before ripening and the end of fruit growth. The last time-point for this analysis was at harvest. Analysis was done as described [38] with safranin/fast green staining.

### Oil drops—histological analysis

A fresh section of the fruits was analyzed for oil drop size and density using Sudan IV staining protocol. The specimens were stained for 6 min with Sudan IV (0.5% w/v in 90% ethanol) and then transferred to a slide, differentiated rapidly in ethanol 50% to remove excess stain and the images observed and photographed under a light microscope DMLB (Leica, Germany) with a DS-Fi1 camera attached (Nikon, USA). Measurement of oil drop size and density was done using NIS elements software (Nikon, USA).

### Cold-press—olive oil extraction

At the end of the experiment, olives of each cultivar and location were harvested after developing a semi-black skin color, and a maturity index (MI (of approximately three, which occurred around November. The MI was calculated from three repeats of 100 fruits, as a subjective evaluation of the skin color and flesh as developed in the Research Station of Venta del Llano (Jaen, Spain) and proposed by Uceda and Frias [39]. Oil was extracted from healthy fruits using a laboratory-scale Abencor system (Comercial Abengoa, S.A., Seville, Spain) equipped with a hammer crusher, malaxer and centrifuge that simulates the industrial process of EVOO production.

### Determination of phenolic compounds

In 2016, Phenolic compounds were isolated from a solution of oil in hexane by double-extraction with methanol/water (60:40, v/v). Total phenols, expressed as tyrosol equivalents (ppm), were determined with a UV–visible spectrophotometer (Beckman Coulter, Fullerton, CA, USA) at 735 nm using Folin–Ciocalteu reagent.

In 2017, the phenolic compounds were isolated from olive oil to evaluate Ortho-diphenols by solid-phase extraction as developed by Mateos et al., [40]. Determination of biophenols by HPLC was done according to International Olive Council (COI/T.20/Doc No 29/Rev.1). The calculation of Biophenol content, expressed in mg/kg, was calculated by measuring the sum of the areas of the related chromatographic peaks.

### Determination of oil fatty acid composition

Fatty acids were transformed into fatty acid methyl esters (FAMEs (using trans-esterification with cold methanolic solution of potassium hydroxide, according to International Olive Council (COI/T.20/Doc. 24) and European Union (EU Regulation- EN 1991R2568) protocols. Data analyzed by Chemstation software.

The calculation of the percentage of each fatty acid identified by gas chromatograph was done according to the formula: %fatty acid = (area of fatty acid x 100) / (total area)

### Statistical analysis

The parameters tested were fresh and dry fruit weight, dry fruit oil concentration, wet oil concentration, amount of oil per fruit, cell size and number of cell layers and oil droplet size and

density. These were subjected to three-way analysis of variance (ANOVA) including full factorial analysis for each year, for their dependence on the three independent variables of sampling date, tree location and cultivar type, including the various interactions between them.

Fruit was sampled during the entire period from fruit set to ripening. Obviously, fruit growth and oil accumulation continued throughout this period. Therefore, we chose two sampling days and performed a full factorial two-way ANOVA analysis for the independent variables of tree location and cultivar type for each. The first sampling date chosen (146 days post-anthesis), was that on which the differences between the two locations were most evident. Harvesting time was chosen as the obvious summation date of the experiment. When we encountered significant interaction between factors, a Tukey-Kramer test was performed in order to rank the various levels of interaction. All statistical analyses were performed using JMP software [41].

## Results

### Different locations typify different climate conditions

Summer temperatures in Tirat Zvi (the HT site) were higher than in Tzuba (the MT site) by almost 10°C in daytime and 5°C at night. For example, in 2016 the average daily maximum temperature at the HT site was above 40°C during June, July August and September, whereas at the MT site it was 32.8, 32.8, 32.6 and 30.8°C respectively during these months. During this period, there was only one day with a temperature above 40°C in the MT site (40.6°C). The average daily minimum temperature at the HT site was above 21°C from June to September, whereas in the MT site it never exceeded 20°C. During 2017, the maximum monthly temperature was above 40°C in the HT site from May till October. During November and December, temperatures decreased dramatically at the HT site. During July, the maximum temperature at the HT site was 45.8°C whereas at the MT site it was 37.8°C. In 2017, the average daily minimum temperature in the HT site was above 20.5°C from June to September, and below 20.5°C in the MT site. The average difference in maximum daily temperature between the HT site and the MT site was 9.2°C and 7.5°C during 2016 and 2017 respectively and a difference of 4°C and 4.15°C in the average minimum daily temperatures during 2016 and 2017 respectively (Fig 1 and S1 Table). During both seasons, there were only few rainy days. The humidity at the MT site was slightly higher compared to the HT site and wind speed in both locations differed in an average speed of 0.47 m/s (S2 Table). Since all trees were kept together at the volcani center until fruit set, differences in flowering time of the various cultivars were not recorded. In both years, fruits were harvested at a maturity index of approximately 3. In 2016, the time of ripening of the fruits was uniform for all cultivars at both locations and the harvest was carried out at the beginning of November with no significant differences between maturity index at the time of harvest for all cultivars at both locations. In 2017, fruits at the MT site ripened earlier than at the HT site. In contrast to the MT site, differences in ripening time between cultivars were observed at the HT site. Therefore, in the MT site, all cultivars were harvested at 168 days post anthesis (October 23rd). In the HT site, the cultivars 'Koroneiki', 'Picholine' and 'Souri' were harvested 198 days post anthesis (November 22nd) and 'Barnea' and 'Coratina' were harvested 226 days post anthesis (December 20th). The maturity index for the various cultivars is presented in S3 Table.

### A high temperature environment affects fruit weight and oil accumulation

The difference between summer temperatures at the HT site and the MT site was greater in 2016 compared to 2017. Therefore, differences between the two locations in all parameters measured were greater in 2016. Interaction between sampling date, cultivar type and tree

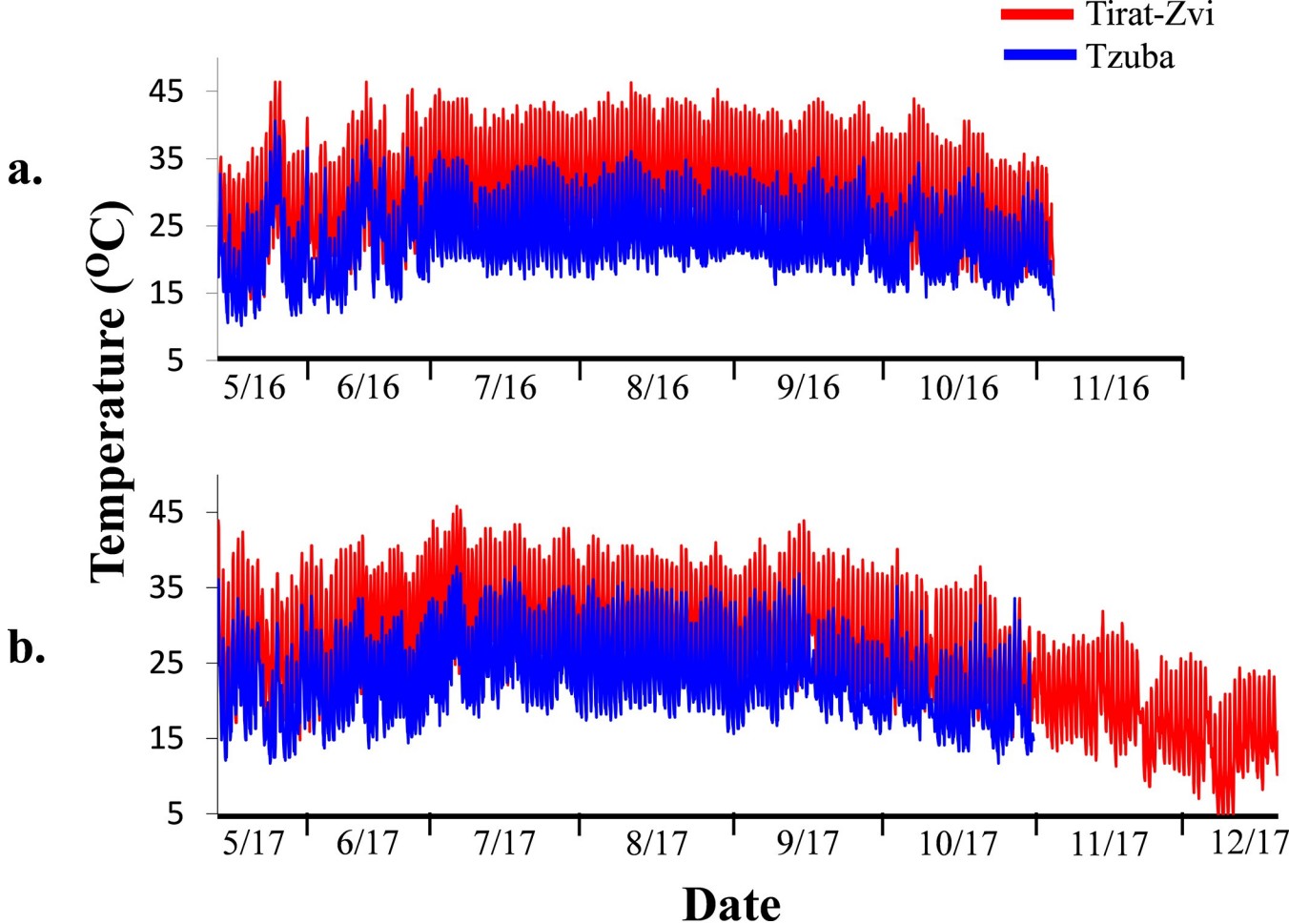

**Fig 1.** Temperatures measured at Tirat Zvi, the HT site (red line), indicate a very warm summer and Tzuba, the MT site, (blue line), represents a mild summer, during 2016 (a) and 2017 (b). Temperature was measured by a data logger every 2 hours during the entire fruit development period from fruit set till harvest.

location was found to be significant for dry fruit weight as well as dry fruit oil concentration (Three-ways ANOVA—P<0.001 and P<0.005 in 2016 and P<0.001 and P<0.001 in 2017 respectively). We tested the effects of cultivar type and tree location and the interaction between them on dry fruit weight and dry fruit oil concentration, by two-ways ANOVA at 164 DPA and at harvest in 2016 and at 146 DPA and harvest time in 2017 (S4A and S4F Table). These independent variables and the interaction between them were found to have significant effects on both parameters at both sampling dates for both years.

'Barnea' dry fruit weight in August 2016 was significantly lower in the HT site trees than that measured at the MT site. In 2017, dry fruit weight of 'Barnea' in the HT site was significantly lower from August till the end of October (harvest time at the MT site). However, when temperatures decreased in November-December, 'Barnea' dry fruit weight at the HT site rose to the same level as at the MT site.

'Koroneiki' dry fruit weight during 2016 was significantly higher at the MT site compared to the HT site, since its dry fruit weight at the HT remained constant from July till harvest. In 2017, dry fruit weight was equal at both locations from fruit set till the end of July. However, during August and September 'Koroneiki' fruit weight increased dramatically at the MT site, while at the HT site fruit weight was roughly constant during this period. At harvest,

'Koroneiki' fruit weight at the MT site was significantly higher than at the HT site (405.3 and 254 mg respectively).

'Coratina' dry fruit weight was equal at the HT and MT sites until July, but after July of both years, there was a significant rise in dry weight at the MT compare to the HT site. At the 2016 harvest, dry fruit weight was double at the MT site compared to the HT site. In 2017, at the MT harvest time, 'Coratina' dry fruit weight at the MT site was 1.38 g, and at the HT site, 0.871. However, when temperatures decreased in November-December, fruit weight at the HT site rose and at HT harvest time reached 1.12 g, still significantly lower compared to the MT 'Coratina' dry fruit weight at harvest. 'Souri' cultivar dry fruit weight was significantly higher at the MT site from the end of July till harvest during both years.

'Picholine' dry fruit weight was significantly higher at the MT site than at the HT site, from August until harvest. During harvest, 'Picholine' dry fruit weight was 1.66 g at the MT site, significantly higher than 1.28 g at the HT site (Fig 2). Fresh fruit weight, commercial oil percentage and oil per fruit data are presented in S5 Table.

The 'Barnea' dry fruit oil content was not significantly different between the two locations in 2016 except in August when dry fruit oil percentage was 30.6 at the MT site and only 16.3 at the HT site. In 2017, beginning in September, dry fruit oil content was higher in the MT trees compared to the HT site (34% and 23.1% respectively). However, since the HT 'Barnea' was harvested much later than in the MT, the final dry fruit oil content at harvest was similar at both locations. The dry fruit oil content in the 'Koroneiki' cultivar was significantly higher at the MT compared to the HT site during both years from August until harvest. The oil percentage of dry fruit of the 'Coratina' cultivar was significantly higher at the MT site compared to the HT site during most of the season. However, in both years, at harvest time, the dry fruit oil percentage was similar at the MT and HT sites (43.6 and 40% in 2016, 46.3 and 44.8% in 2017 respectively). From August till harvest time, the dry fruit oil percentage of the 'Souri' cultivar was significantly higher at MT compared to the HT site during both years. The dry fruit oil percentage of the 'Picholine' cultivar was significantly higher at the MT compared to the HT site during August and September. However, at harvest, the dry fruit oil percentage in the MT and HT sites were similar (32.7% and 30.6% respectively) (Fig 2).

We analyzed the variation in the gain of dry fruit weight and dry fruit oil concentration in all the cultivars tested, during each month from June to September at the two locations in 2016 and 2017. We found that the main disparity in gain of fruit weight between the HT and MT sites occurred during August of both years. Similarly, the main difference in oil concentration between the HT and MT sites was measured during July and August of 2016 and in August of 2017. Since the interaction between cultivar type and tree location was significant for both traits, during August (164 DPA in 2016 and 146 DPA in 2017) and at harvest time, we ranked the cultivars by their performance at the MT site compared to that at the HT site (S4B–S4E and S4G–S4J Table). In 2016, at harvest, the, 'Koroneiki' and 'Coratina' cultivars showed the most significant difference in dry fruit weight, whereas the 'Barnea' cultivar exhibited the least difference between the MT and HT sites. A similar analysis for oil content of dry fruit demonstrated that the 'Koroneiki' cultivar showed a highly significant difference between the two locations, whereas variation for this trait in the 'Coratina' and 'Barnea' cultivars was significantly lower. In 2017, at harvest, the 'Souri' cultivar showed the most significant difference in dry fruit weight between the MT and HT sites. The cultivars 'Koroneiki', 'Picholine' and 'Coratina' exhibited lesser differences between sites, whereas the 'Barnea' cultivar exhibited the least difference between the MT and HT sites. A similar analysis for oil content of dry fruit demonstrated that the 'Koroneiki' and 'Souri' cultivars showed a highly significant difference between the two locations, while the cultivars 'Picholine', 'Coratina' and 'Barnea' had significantly lower variation between locations. We also analyzed the association between the gain of dry

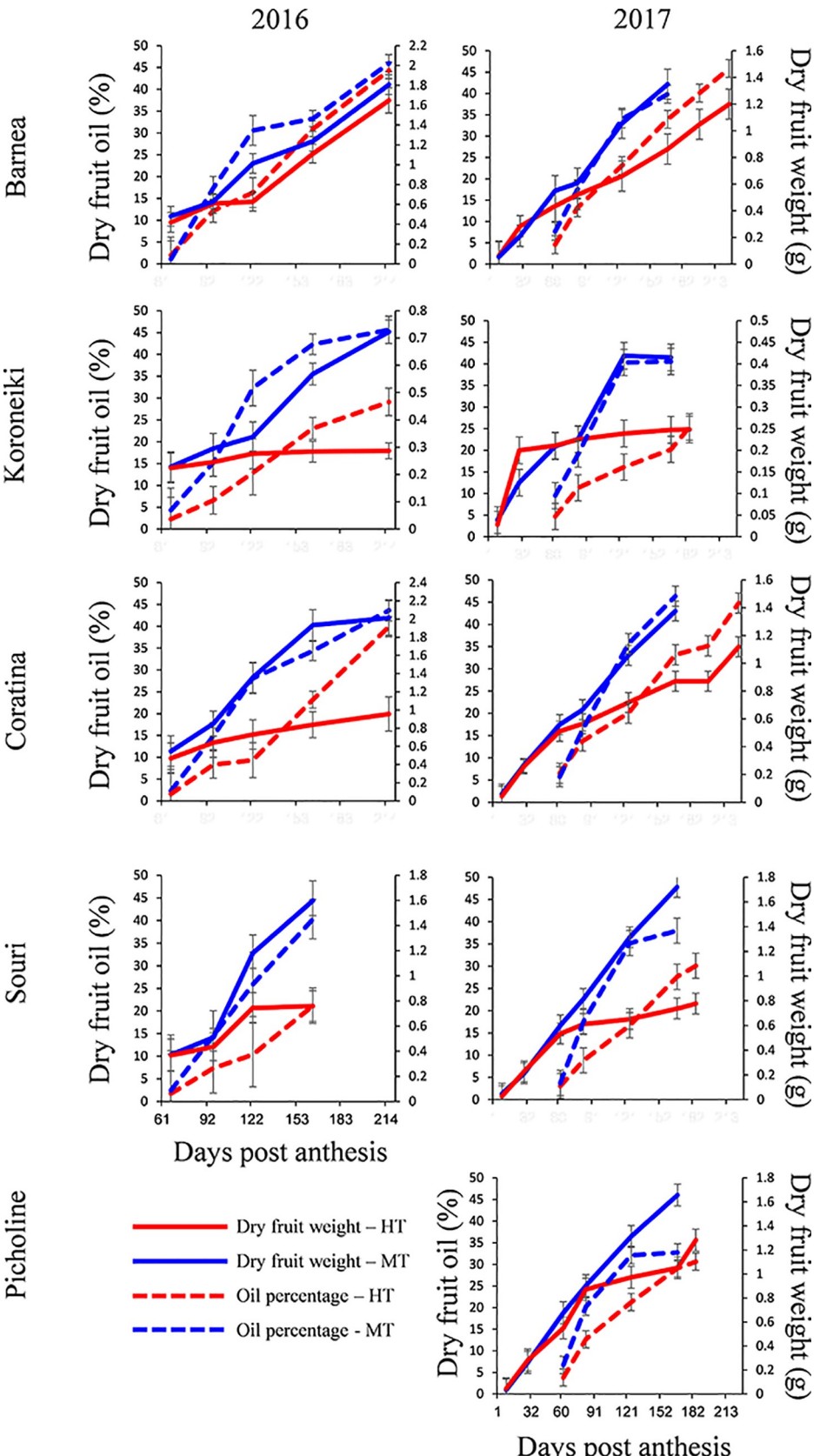

**Fig 2. Dry fruit weight and oil accumulation of all analyzed cultivars during 2016 and 2017.** Dry fruit weight (continuous line) and dry fruit oil percentage (dashed line), of fruits grown at the HT site, Tirat Zvi (red line) and the MT site, Tzuba (blue line), during the entire season are presented. The right Y axis is the dry fruit weight, while the left Y axis is the dry fruit oil percentage. Error bars represent confidence limits ($\alpha$ = 0.05). Error bars that do not overlap represent significant differences.

fruit weight and dry fruit oil concentration in all analyzed cultivars during each month in relation to the monthly average maximum daily temperature (Tmax), minimum daily temperature (Tmin) and average daily temperature (Tmean). We found that dry fruit weight in all cultivars was correlated (negatively and significantly) to all the above temperature variables. In contrast however, dry fruit oil concentration was not correlated to any of those temperature variables monitored. When we applied these analyses to those cultivars in which dry fruit weight was sensitive to high temperatures ('Koroneiki', 'Coratina', 'Souri' and 'Picholine'), we found that gain of dry fruit weight exhibited a significantly negative correlation with all temperature variables. Similarly, when we applied our analysis to those cultivars in which oil concentration had proved itself to be sensitive to high temperatures ('Koroneiki' and 'Souri'), we found a significant negative correlation between dry fruit oil concentration and Tmax (S6 Table).

## High temperature environment affects mesocarp growth

In order to address the effect of a high temperature environment on fruit growth, we measured mesocarp cell size and the number of cell layers in the mesocarp at various developmental stages (Fig 3). The effect of cultivar type and tree location on mesocarp cell size and the number of cell layers in the mesocarp was found to be significant at both 146 DPA and harvest time. The interaction between cultivar type and tree location was found to have a significant effect on cell size at both sampling dates, but affected the number of cell layers at harvest time alone (S4F Table). The 'Barnea' fruit weight curve at the MT site exhibits the expected double sigmoid pattern, in which fruit weight increases linearly after fruit set, stays steady during pit hardening and increases again after pit hardening until September, just before ripening. At the HT site, the 'Barnea' fruit weight curve increases in approximately linear fashion from fruit set to harvest (Fig 2). The number of cell layers in the 'Barnea' fruits at the MT site did not change statistically during fruit growth from the beginning of June until harvest-time at the end of October. However, cell size increased dramatically from 1181 $\mu m^2$ at the beginning of June to 6604 $\mu m^2$ at the end of October. The number of cell layers in 'Barnea' fruits at the HT site was constant from June till September. However, it increased from 27 to 37 layers from September until harvest time in December. The cell size curve of the 'Barnea' fruits at the HT site showed the opposite tendency and increased from June till September but remained steady from September until December (Fig 3 and S2A Fig).

The 'Koroneiki' fruit weight curve at the MT and HT sites is similar to that of 'Barnea'. At the MT site it exhibits double sigmoid growth and at the HT site growth is linear. However, the slope of the 'Koroneiki' curve at the HT site is flatter than in the 'Barnea'. The number of mesocarp cell layers as well as cell size in the 'Koroneiki' fruits in the MT site increased during the entire period of fruit development. In June, the 'Koroneiki' mesocarp in the MT site consisted of 20 layers of 1596 $\mu m^2$ cells, whereas in October (at harvest) it consisted of 34 layers of 6210 $\mu m^2$ cells. At the HT site, mesocarp cell size and the number of cell layers increased dramatically from June to July, before pit hardening, and then increased at a slower pace till harvest. During this period cell size as well as the number of cell layers was significantly lower in the mesocarp of 'Koroneiki' fruits from the HT site compared to those which grew at the MT site (Fig 3 and S2B Fig). The 'Coratina' mesocarp showed a tendency similar to that of 'Koroneiki', however the number of cell layers, unlike in the 'Koroneiki' mesocarp, was higher at the

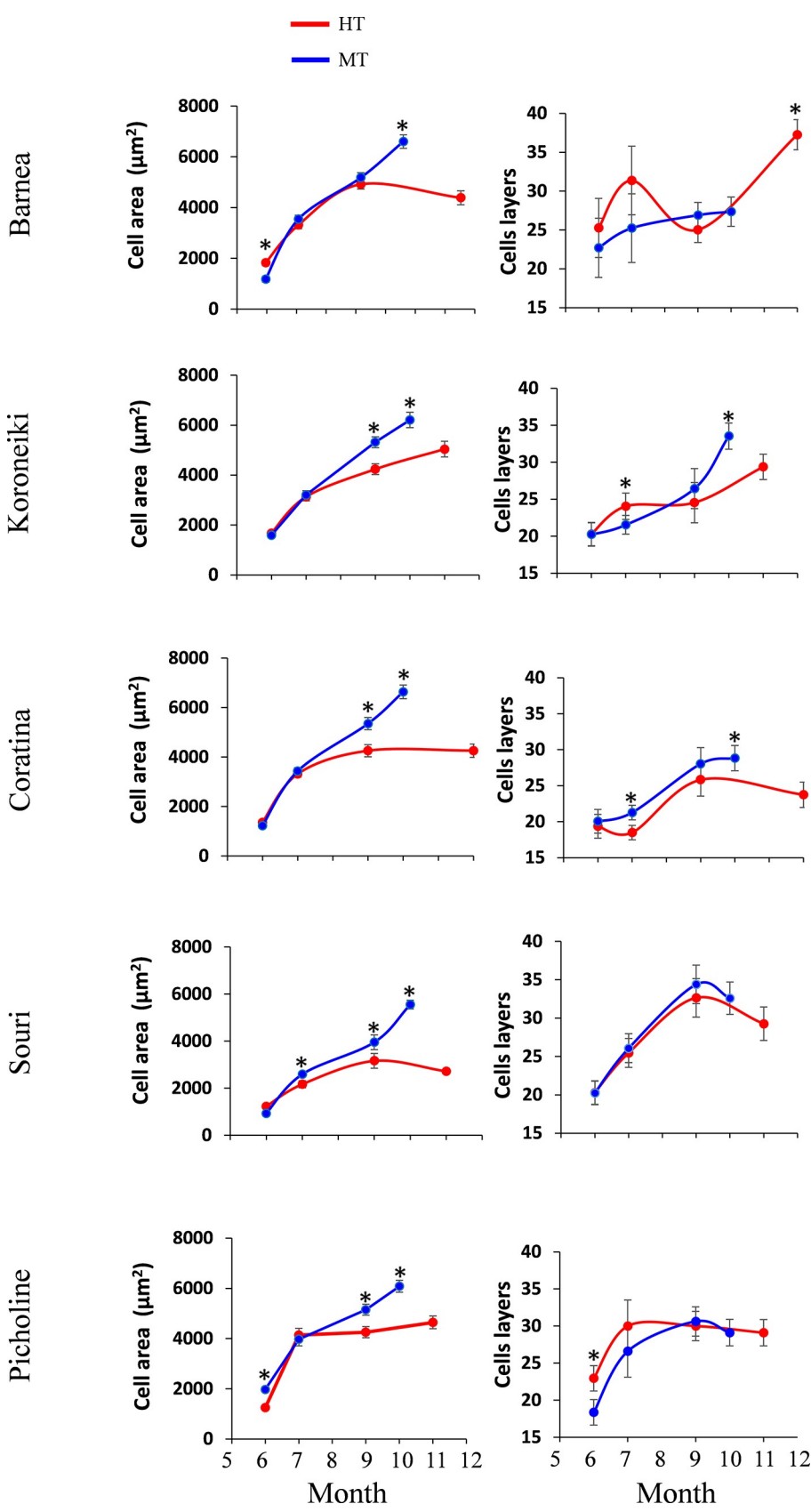

**Fig 3. Mesocarp cell characterization in all 5 cultivars during 2017.** Quantification of the average cell area and the number of cell layers during the season (June, July, September and at harvest) are presented. Error bars represent confidence limits ($\alpha = 0.05$). Asterisks represent significant difference ($\alpha = 0.05$).

MT site all season. The number of cell layers in the 'Souri' and 'Picholine' mesocarp was similar at both sites, whereas cell size in both cultivars was significantly higher at the MT site compared to the HT site in three of the four dates sampled (Fig 3).

September fruit weight of all five cultivars at the MT site was significantly higher than at the HT site. However, at that time, the number of mesocarp cell layers in the fruit of all cultivars at the MT site was not significantly higher than at HT. In contrast to 'Barnea', the mesocarp of the other four cultivars consisted of significantly larger cells in the MT site trees compared to the same cultivars grown at the HT site (S4 Fig).

## A high temperature environment affects oil drop size and density

In order to address the effect of a high temperature environment on oil accumulation, we measured oil drop size and density at various stages of development: 50, 83 and 146 DPA (Fig 4). At 146 DPA, the effects of cultivar type and tree location and their interaction, on oil drop size, was found to be significant. In contrast, the effect of tree location on oil drop density was not found to be significant. However, the effects of cultivar type and the interaction between cultivar type and tree location were significant (S4F Table). 'Barnea' fruit at the HT site, at 50 DPA, showed significantly larger, denser oil drops compared to those at the MT site. Average oil drop size was 75 and 54 $\mu m^2$ and average oil drop densities were 476 and 352 drops per $cm^2$, in the HT and the MT sites respectively. However, later in the season, at 83 as well as 146 DPA, the oil content and oil drop size were higher at the MT compared to the HT site, whereas oil drop density was similar at both locations. Oil drop sizes at 83 DPA were 286 and 220 $\mu m^2$ at the MT and the HT sites respectively and at 146 DPA, 860 and 658 $\mu m^2$. The differences between the two locations on both dates are significant. Oil drop densities at 83 DPA were 747 and 650 drops per $cm^2$ and at 146 DPA, 677 and 645 drops per $cm^2$ at the MT and the HT sites respectively. These differences on both dates are not statistically significant (Figs 4 and S4F). 'Koroneiki' olives showed the same trend as the 'Barnea' with bigger oil drops at the HT site at 50 DPA and bigger oil drops at the MT site at 83 DPA and at 146 DPA. However, at 146 DPA, in 'Barnea', the ratio of oil drop size between the MT and the HT sites is 1.3 and in 'Koroneiki', the ratio reaches 2.6. In addition, at 146 DPA, oil drop density of 'Koroneiki' olives is higher in trees at the HT compared to the MT site (Fig 4 and S4B Fig). 'Coratina' oil drop density was higher at the MT site compared to the HT site. However, the differences between the two sites was not significant at all three sample dates. Surprisingly, 'Coratina' oil drops were significantly larger at the HT site compared to the MT site at 83 DPA. However, later in the season, at 146 DPA, oil drops were significantly larger at the MT site compared to the HT site. The oil drop density in 'Souri' and 'Picholine' was similar at both sites at all sampling dates. However, oil drops were significantly larger at the MT sites compared to the HT sites in both cultivars (Fig 4). By September, the oil content of all five cultivars was significantly higher in olives at the MT site than in those grown in the HT site. At that time, oil drop size was also significantly greater in the MT site olives than in those from the HT site. However, oil drop density in 'Koroneiki' olives was higher at the HT than at the MT site (S5 Fig).

## High temperature environment affects oil composition

In order to characterize the effect of high temperature environments on oil composition and quality, we measured the polyphenol content in the oil extracted from the various cultivars at

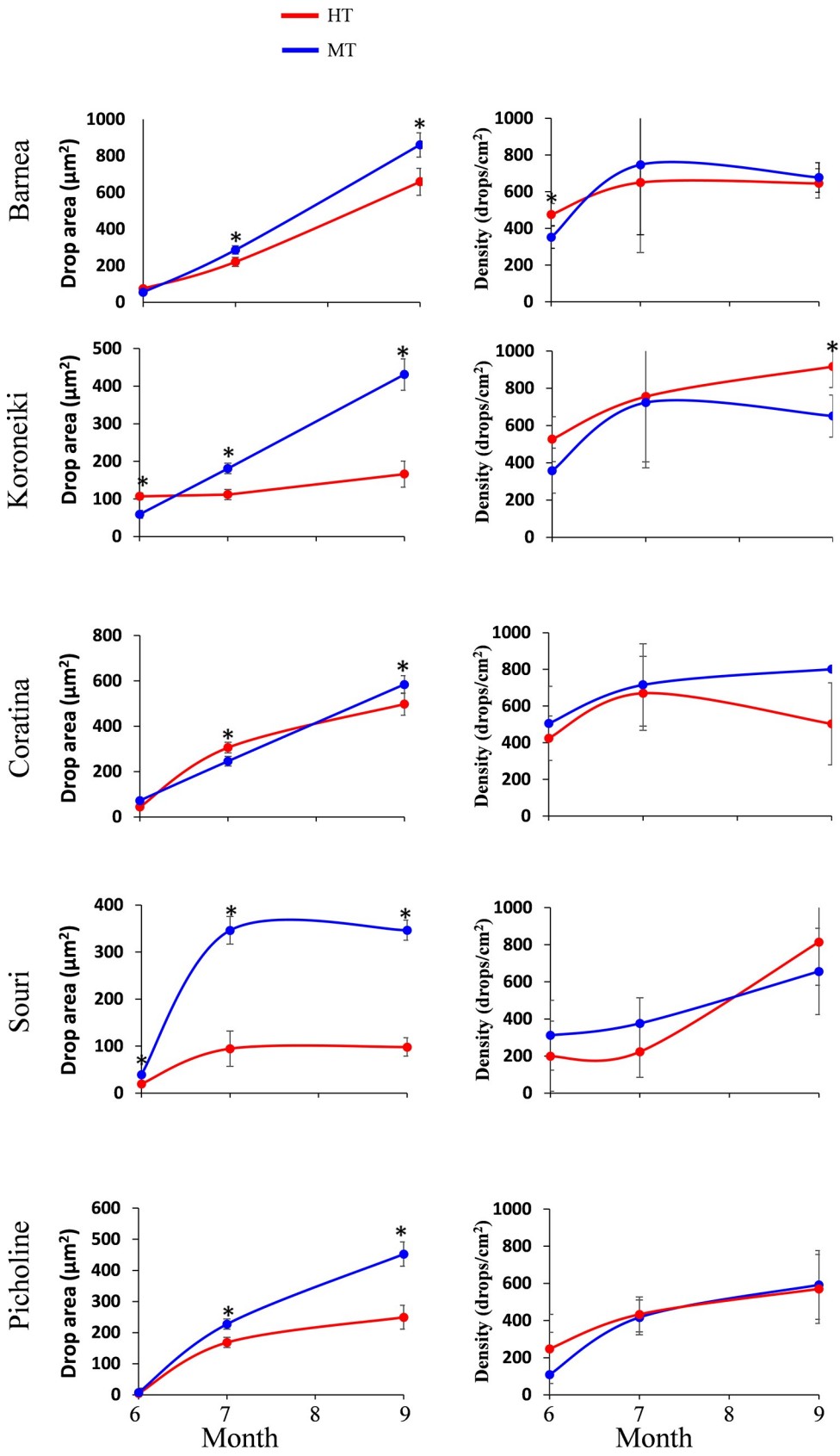

**Fig 4. Oil accumulation in all cultivars during 2017.** Quantification of the average oil drop area, the density of oil drops as well as the oil content during the season (June, July and September) are presented. Error bars represent confidence limits (α = 0.05). Asterisks represent significant difference (α = 0.05).

both locations (Fig 5A). During the 2016 season, polyphenol levels varied between cultivars. In general, levels were almost double in oil extracted from fruits from the MT site compared to those grown at the HT site. In 2017, polyphenol levels in all cultivars at the HT site were lower than at the MT site. The lowest polyphenol level, 156 mg/g oil, was measured in the oil extracted from 'Barnea' from the HT site. The polyphenol level in oil extracted from 'Barnea' grown at the MT site was more than double that from the HT fruits—404 mg/g oil. Oil extracted from 'Coratina' showed the same pattern as that of 'Barnea'. Polyphenol levels in oil extracted from 'Picholine' grown at the MT site were more than three times higher than in oil extracted from 'Picholine' from the HT site (893 and 291 mg/g oil respectively). Polyphenol levels in oil extracted from 'Koroneiki' at the MT site were 150% higher than the levels at the HT site. The smallest difference between polyphenol levels in oil extracted from the two locations was found in oil extracted from the 'Souri' cultivar. This was mainly due to the unusually high level of polyphenols found in oil extracted from 'Souri' grown at the HT site, 772 mg/g oil (Fig 5A).

The main fatty acid found in olive oil is oleic acid. We found that oil extracted from all five cultivars grown at the HT site contained lower levels of oleic acid than oil extracted from olives grown in a milder environment (Fig 5B). During 2016, the differences between sites in the percentage of oleic acid in oil extracted from olives grown at the MT site compared to olives from the HT site was 8, 4 and 7% in 'Barnea', 'Koroneiki' and 'Coratina' respectively. During 2017, oleic acid levels were higher than in 2016 for all cultivars in both climate regions. Oleic acid levels in oil extracted from olives grown at the MT site reached 74% in 'Koroneiki' and 76% in 'Coratina', whereas the levels in oil extracted from olives grown at the HT site was 67 and 69% respectively. Oleic acid level in oil extracted from 'Picholine' olives grown at the HT site was 51.8%, whereas at the MT site it was 60.4% (Fig 5B). We also characterized all other fatty acids in the oil of the various cultivars in both climate regions (S7 Table). The decrease in oleic acid content in the oil extracted from olives grown at the HT site coincides with an increase in the level of palmitic acid (C16:0) and linoleic acid (C18:2) in oil extracted from olives grown there. The palmitic acid content was about 2% more in oil from the HT site compared to the MT olives in all cultivars during both years. Excessively high levels of palmitic and linoleic acid were detected in 'Barnea' oil extracted in 2016, and in 'Picholine' oil extracted in 2017. 'Picholine' oil extracted from olives grown in 2017 in the HT site contained 21.6% palmitic acid, and 'Barnea' oil extracted from olives grown in 2016 at the HT site contained 23.3% linoleic acid.

## Discussion

The most critical stage of development of olive fruit, when fruit growth as well as oil accumulation rate are maximized, is between 60 and 120 days after flowering, between pit hardening and fruit ripening [42]. In the northern hemisphere, this period falls in July and August. In both 2016 and 2017, this period was particularly hot at the HT site. Temperatures reached 46°C in 2016 and 45°C in 2017, whereas the mean daily maximum temperatures during July and August were 42.5°C in 2016 and 40.4°C in 2017. In general, the summer of 2016 at the HT site was hotter than the summer of 2017. This may be the reason for the early ripening in 2016 at the HT site compared to 2017. However, it was shown that several parameters such as irrigation regime or fruit size can affect ripening time [43].

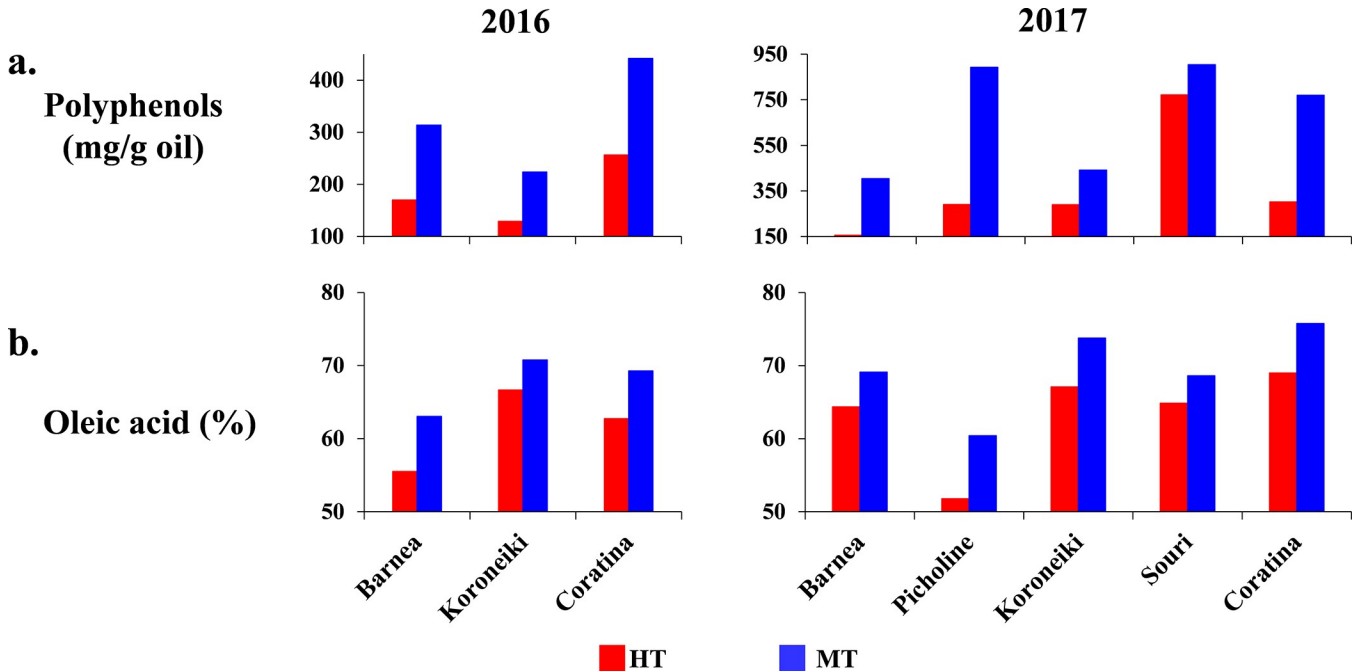

**Fig 5.** Oil quality, represented by polyphenol (a) and oleic acid (b) levels in the oil extracted from the various cultivars in the two climatic regions, during both years.

Olive oil yield is the function of the number of fruits, average fruit weight and oil concentration at maturity. In this study, we addressed the effects of an environment of consistently high temperatures on fruit development, oil accumulation and oil composition in five olive cultivars. We found that different olive cultivars respond to high temperatures of the environment in a genotypic specific manner. A high temperature environment repressed fruit development and oil accumulation and also affected oil composition. We found that the 'Koroneiki' and 'Souri' cultivars are highly temperatures sensitive, and were affected to an extreme degree by the high environmental temperatures. We demonstrated that fruit weight and oil accumulation in 'Koroneiki' and 'Souri' olives harvested in the HT site were much lower compared to these parameters at the MT site. In contrast to 'Koroneiki' and 'Souri', the 'Barnea' cultivar exhibited greater tolerance to high temperatures, and although fruit growth as well as oil accumulation were at some stages significantly lower at the HT site, the final fruit weight as well as oil percentage at harvest was virtually the same at both the HT and MT sites (S4 Table). These results were repeated in 2016 and 2017 (Fig 2). Oil quality, defined by oleic acid content as well as by polyphenol levels in the oil after harvest, was found to be affected by high temperatures in all five cultivars.

Whether the effects of high temperatures can be attributed mainly to average, maximum or minimum daily temperatures or to other temperature parameters, is still not clear. We found significant association between dry fruit weight and Tmax, Tmin and Tmean. However, when we analyzed sensitive cultivars only, we found a significant association between Tmax and dry fruit oil concentration (S6 Table). Daily maximum temperature differences between the HT and the MT site were higher in 2016 compared to 2017. On the other hand, differences between the HT and MT site in the daily minimum temperatures were higher in 2017. Differences in dry fruit weight and dry fruit oil concentration were also higher in 2016. It would seem therefore, that the most relevant parameter for predicting the effects of high

temperatures, is the daily maximum temperature. A recent study [35] found that olive fruit dry weight showed a tendency to decrease with increasing mean temperature, while the proportion of oil in the fruit exhibited a significant correlation with mean daily thermal amplitude, and weaker correlation with mean daily maximum and minimum temperatures. The proportion of oleic acid in the oil showed a negative correlation with mean daily minimum temperature and with mean daily temperature, and a linear relationship with mean daily thermal amplitude [35]. This study used lower maximum temperatures to cause heat stress compared to the ambient temperatures at our HT site (S1 Table). We can therefore surmise that olive plants suffered from high temperatures as a result of the maximum daily temperatures encountered at the HT site.

## A high temperature environment affects fruit weight

We found that final dry fruit weight at harvest of 'Barnea' cultivar was not affected by the location of the trees nor by the resulting difference in temperatures. In comparison to 'Barnea', the 'Koroneiki', 'Coratina', 'Souri' and 'Picholine' cultivars showed a decreased dry fruit weight at harvest in response to the higher temperatures at the HT site. The influence of temperature during early stages of fruit development has been identified in several fruit tree species [44–47]. Dry fruit weight in the olive cultivar 'Arauco' was not affected by temperatures in the range of 16–25°C. However, at higher temperatures it showed a decrease of 0.08 grams in dry fruit weight for each additional 1°C [30]. We found a cultivar dependent decrease of between 0.026 to 0.044 grams in 2016, and 0.015 to 0.078 grams in 2017, in dry fruit weight per 1°C increase, measured in September, before harvest.

It has been suggested that temperature during fruit development may have a greater effect on cell division than its effect on cell expansion in tomato fruits [48]. Another study found that continuous heating of tomato fruit reduced cell expansion [49]. We found that in 'Barnea' olives grown at the MT site, cell expansion alone contributed to mesocarp growth beginning 7 weeks after full bloom. At the HT site, 'Barnea' fruits followed this pattern from 50 to 146 days (7 to 21 weeks) after full bloom. However, during the period of 146 to 247 days after full bloom, cell expansion ceased and cell division alone contributed to mesocarp growth. In the 'Koroneiki' cultivar we found a similar pattern to 'Barnea', the exception being, that at the MT site, cell division also contributed to mesocarp growth in the late stages of fruit development. 'Coratina' showed the same trend as 'Koroneiki' whereas in 'Souri' and 'Picholine', as in 'Barnea', it was cell expansion alone that was responsible for the differences between the two sites (Fig 3). Rallo and Rapoport [19] found that in olives, as in other drupes, both cell division and cell expansion contribute to mesocarp growth at early stages of fruit development. Six weeks after full bloom, cell division ceased, and cell expansion alone contributed to further mesocarp growth. Hammami et al. [20] found that fruit size differences among six olive cultivars are due to cell division throughout fruit growth, which occurs mainly in the first six weeks after bloom. However, they were surprised to find that a substantial number of cells formed after these six weeks, and continued during the following 6 months. The six cultivars they tested did not include the five cultivars tested in this study. In our study, it is possible that in reaction to the unusually high temperatures at the HT site during the early stages of fruit development, the "Barnea" cultivar delayed mesocarp cell division until the arrival of milder weather (October-December). The 'Koroneiki' cultivar, which has been shown to be relatively sensitive to high temperatures, may have been stressed by the temperatures at the MT site during the spring and as a result, exhibited delayed cell division. In September 2017, when fruit weight differences between olives grown at the HT site and those of the MT site were at their greatest, the number of cell layers was identical, but cell area was significantly greater at the MT site in

all cultivars but the 'Barnea'. This suggests that unlike satsuma mandarins [47], the main effect of high temperatures on olive fruit weight is through repression of cell expansion and not cell division (S3 Fig).

## High temperature environment affects oil accumulation

We found that the effect of a high temperatures environment on oil concentration is genotype dependent. In 2016, the average difference in maximum daily temperature between the HT and the MT site was 9.2˚C, and that of minimum daily temperature was 4˚C. In 2016 these temperature differences did not affect 'Barnea' and 'Coratina's' final oil concentration. However, the 'Koroneiki' cultivar of the same year had a final dry fruit oil concentration at harvest of 45.6% at the MT site and 29.1% at the HT site, which is an average of about a 1.79% decrease per degree of increased maximum daily temperature and 4.13% decrease per degree of increased minimum daily temperature. The 'Souri' cultivar had a 2.08% decrease per degree of increased maximum daily temperature and a 4.78% decrease per degree of increased minimum daily temperature. In 2017, decrease in oil concentration as a function of the severity of high temperatures, was more moderate than in 2016. However, just as in 2016, dry fruit oil concentration of 'Barnea' and 'Coratina' cultivars was not affected by the location of the trees and the resulting difference in temperatures, whereas the 'Koroneiki' and 'Souri' cultivars showed a decreased oil concentration at harvest in response to the higher temperatures at the HT location. 'Picholine' dry fruit oil concentration, like that of 'Barnea' and 'Coratina', was similar at both locations. Other studies have found that heat stress reduced oil concentration in sunflower hybrids by 6% [50]. However, corn which had undergone heat stress during grain fill, was found to have the same oil content as control plants [51]. Olive oil concentration as a percentage of dry weight in 'Arauco' cultivars was found to decrease linearly at 1.1% per degree of increased temperature. [30]. Our finding is also consistent with the results of Rondanini et al. [33] which measured the oil accumulation of six olive cultivars, at three locations over two years and found a negative association between oil concentration and average temperature. The response to heat stress treatment of potted 'Coratina' and 'Arbequina' trees was found to reduce dry fruit weight by 0.34 and 0.22 g, respectively. In this study, fruit oil concentration (%) was 4.6 and 6.2% less on a dry-weight basis in fruit exposed to elevated temperatures. Higher temperature was found to promote vegetative growth but negatively affected oil concentration [32]. Trentacoste et al. [31] found that fruit oil concentration in 10 olive varieties decreased with increasing maximum daily temperature. It is accepted that oil accumulation begins only after pit hardening. However, while pit hardening occurs about 10 weeks after flowering, Matteucci et al. [28] has shown that oil bodies appear in the mesocarp cells 7 weeks after flowering, and large oil droplets, derived from their fusion, are present in each cell 10 weeks after flowering. In accord with Matteucci et al. [28], we also observed very clear oil bodies 50 days after flowering. In September, 146 days after flowering, oil drops were significantly larger at the MT site compared to the HT site in all five cultivars. At this stage, oil drops comprise between 3% to 17% of the cell volume. However, these values do not represent the true oil concentration in the mesocarp, because each cell may include many small oil drops that had not yet fused at the time the picture was taken.

## Differences between years of the experiment

Although development patterns of the olive fruits were similar during both years of the study, the differences between those grown at the MT site and those at the HT site, were significant. Humidity, wind speed and rainfall were similar in 2016 and 2017 (S2 Table). However, the yearly average temperature difference between the HT and the MT site was higher in 2016

compared to 2017. Another result of our study indicates that during the course of the growing season, August is the month in which temperature has the greatest effect on fruit weight and oil accumulation, since during this month we found the greatest divergence between the two locations in 2016 and 2017 (S6 Table).

## High temperatures environment affects oil quality

Among the effects of high temperatures on chemical parameters characterized in the current study, were total polyphenols and the fatty acid profile of olive oil. Polyphenol levels are known to decrease during fruit development [52, 53]. These levels are often positively correlated with water stress but in some cases show a divergent trend [54, 55]. Our analysis shows that a high temperature environment caused a decrease in the total polyphenol content of all analyzed cultivars. In 2016, total polyphenol level of oil from the HT site was approximately 55% of the level in oil from the MT site in all 3 analyzed cultivars. In 2017, total polyphenol level oil from the HT site was approximately 35% the level found in oil from the MT site in 'Barnea', 'Picholine' and 'Coratina'. However, in the 'Koroneiki' cultivar, polyphenol level at the HT site was 65% the level in the MT site, and in 'Souri', polyphenol level was very high in the HT site olives, reaching 772 mg/kg oil, compared to 905 mg/kg oil in the MT site. The decrease in total polyphenols in the HT site may also be explained by the differences in the irrigation regime. It is recognized that excess irrigation during fruit development results in lower phenolic concentrations in the olive oil due to changes in the biosynthetic and catabolic polyphenol pathways in the olive fruit [56–60].

The olive trees in the HT site got one third more water than those in the MT site; In our opinion, the difference between polyphenol levels at the two locations are too dramatic to be explained only by the level of irrigation [55]. The total polyphenol levels in 2017 were higher than in 2016 and were very high compared to levels found in previous studies [55]. This can be explained as an effect of climate change. Total polyphenol levels were dramatically lower in oil from the HT compared to the MT site in all cultivars except for the 'Souri', which as mentioned above, had a very high total polyphenol level at the MT site as well as at the HT. This might suggest that, at least in terms of oil quality, the 'Souri' cultivar is relatively tolerant to high temperatures.

One of the major criteria of oil quality is its fatty acid composition. In accord with other studies on olive oil, in our study all cultivars showed a reduction in their oleic acid content in both years in response to the high temperature environment at the HT site. In sunflower oil, heat stress caused an increase in oleic acid content and a reduction of linoleic acid [50]. In contrast, it has been demonstrated that in the olive, high temperatures caused a decreased level of oleic acid and an increase of linoleic and palmitic acids [30]. This may be partly explained by the gene expression level of genes involved in the oil biosynthesis pathway [61]. The cultivars 'Arbequina', 'Barnea', 'Koroneiki', 'Manzanillo' and 'Picual', showed lower level of oleic acid and higher level of linoleic acid in regions with high temperatures compared to the levels found in areas of mild temperatures [37]. This study compared olive oil composition in Northern New South Wales and Southern Queensland, which is a warm area to Tasmania, a milder region. The 'Koroneiki' fatty acid composition was not analysed in Tasmania. However, the 'Barnea' and 'Coratina' showed a decrease in oleic acid content of 16.6 and 7.8% respectively in the warm region compared to the mild region. Our results showed smaller differences between oil extracted from the HT compare to the MT sites in oleic acid content. The temperatures during the 2 season analysed by Mailer et al. [37] were not mentioned. However, the larger decrease in oleic acid content compared to our findings can be explained by larger differences in temperatures or in other agronomic parameters, since the sites in their experiment differed

in many variables: they were cultivated by different farmers, possibly with different soil, water quality and many other parameters. The influence of temperatures during the growing season on the fatty acids composition of 188 Italian cultivars was studied between 2001 and 2005. Significantly lower oleic acid and higher palmitic and linoleic acids levels were found in the warmest year of the study (2003) compared to the coolest year (2005) [36].

The cultivars used in our study all exhibited, in both years of the trial, elevated levels of palmitic as well as linoleic acids as a result of exposure to the high temperatures of the HT site. According to the International Olive Council (IOC; http://www.internationaloliveoil.org), olive oil must contain 55–83% oleic acid and 3.5–21% linoleic acid. The oil of the 'Barnea' grown at the HT site in 2016 contained 23.34% linoleic acid and 'Picholine' oil from the HT site in 2017 contained 51.8% oleic acid and 21.62% linoleic acid. Thus, neither meets the standards of the IOC. Analysis of the oils extracted from the five cultivars in 2017, showed 'Souri' to have the smallest disparity in its oil composition between olives grown at the MT site compared to those from the HT site. Oleic acid in 'Souri' decreased by less than 4% and linoleic acid increased by only 1.5% at the HT compared to the MT site. These results, taken with those of total polyphenol levels, suggest that in regard to oil quality, the 'Souri' cultivar is more tolerant to high temperature environments than any of the other cultivars analyzed in this study.

## Conclusions

Our study demonstrates the negative effect of a high temperature environment on several key characteristics critical to the quality of olive oil. High temperature environments are shown to negatively influence fruit development as well as oil accumulation and thereby reduce yield. These high temperatures diminish oil quality by modifying fatty acid composition and causing a reduction of polyphenols and oleic acid, the most important components of olive oil. However, high temperature effects are genotype dependent and each cultivar responds differently to this stress. We found that each of the tested cultivars responded differently to high temperatures environment; none were completely heat tolerant. The 'Koroneiki' cultivar was negatively affected by high temperatures in all analyzed parameters. In the 'Picholine' and 'Coratina' cultivars, fruit development and oil quality were negatively affected by the high temperatures of the HT site, but oil concentration remained unaffected. The 'Souri' cultivar responded negatively to the high temperature environment with regard to fruit development and oil concentration, but was relatively tolerant to high temperatures in terms of oil quality. In contrast, in the 'Barnea' cultivar, exposure to high temperatures reduced oil quality, but did not impair final fruit weight or oil concentration. Although our results should be treated cautiously since the experiment was carried out for only two years, we found that the tested cultivars were tolerant regarding the effect of a high temperature environment on some of the traits examined, while exhibiting sensitivity to others. This suggests that the response to a high temperature environment is different for each of these traits and indicates induction of three different signal transduction mechanisms, resulting in a reduction of fruit development, oil accumulation and oil quality. Different olive cultivars have developed a variety of mechanisms to deal with different aspects of high temperature damage. Elucidation of the mechanism of each of these three responses is a vital step in the process of developing a variety of olives tolerant to high temperature damage.

## Supporting information

**S1 Fig. Microscope image of 'Barnea' fruit in June 2017.** The region of the mesocarp cell layers counted is marked.
(PPTX)

**S2 Fig.** Microscope images of the mesocarp cells of 'Barnea' (a) and 'Koroneiki' (b) sampled in June, July, September and at harvest-time of the 2017 season from Tirat Zvi (HT site) and Tzuba (MT site).
(PPTX)

**S3 Fig. Mesocarp cell characterization in all five cultivars in September 2017.** Microscope images of the mesocarp cells sampled in September are presented at the left with quantification of the average cell area and the number of cell layer as well as fruit weight at the time, are presented at the right. Error bars represent confidence limits ($\alpha = 0.05$). Asterisks represent significant difference ($\alpha = 0.05$).
(PPTX)

**S4 Fig.** Oil accumulation in 'Barnea' (a) and 'Koroneiki' (b) cultivars during 2017. Microscope images of the mesocarp cells sampled in June, July and September of 2017 season from Tirat Zvi (HT site) and Tzuba (MT site).
(PPTX)

**S5 Fig. Oil content in the mesocarp in all five cultivars in September 2017.** Microscope images of oil drops in fruits sampled at September are presented at the left and quantification of the average oil drop area, the density of oil drops as well as the oil content in September 2017 are presented at the right. Error bars represent confidence limits ($\alpha = 0.05$). Asterisks represent significant difference ($\alpha = 0.05$).
(PPTX)

**S1 Table. Maximum and average of the daily maximum and minimum temperature at both locations for each month, in years 2016 and 2017**
(DOCX)

**S2 Table. Climate data on both seasons at both locations.**
(DOCX)

**S3 Table. Maturity index at harvest of all cultivars at both locations and years.**
(DOCX)

**S4 Table. Two-way ANOVA test. a., f.** The probability of the effects of cultivar type and tree location and their interaction, on the various parameters measured in this study, at two sampling dates in each year, 164 DPA and at harvest in 2016 and 146 DPA and at harvest time in 2017. Significant effects (P<0.05) are in bold font. **b-e, g-j.** Tukey-Kramer test, ranking the differences in dry fruit weight and dry fruit oil content between locations in the tested cultivars in 2016 at 164 DPA (b-c) and at harvest (d-e) and in 2017, at 146 DPA (g-h) and at harvest (i-j). The values appearing in the tables b-e and g-j are the added value in performance at the MT site compared to the HT site, calculated as a proportion of the HT results.
(DOCX)

**S5 Table. Fruit weight, oil percentage and oil per fruit in all cultivars during 2016 and 2017.** Errors are confidence limits ($\alpha = 0.05$).
(DOCX)

**S6 Table. Correlation between Tmax, Tmean and Tmin to dry fruit weight and dry fruit oil percentage.** The monthly gain of dry fruit weight and dry fruit oil percentatge for each cultivar, at each site for both years and was correlated to the temperatures data for each month (average daily maximum temp.–Tmax; average daily temp.–Tmean; average daily minimum

temp.–Tmin). Correlation coefficient was calculated for all cultivars or for heat sensitive culti-vars only. Significant correlation coefficients are highlighted in red font.
(DOCX)

**S7 Table. Olive oil composition in both climate regions during 2016 and 2017.**
(DOCX)

## Acknowledgments

We thank Yehuda Ben-Ari for valuable assistance in writing and editing this paper.

## Author Contributions

**Data curation:** Yael Nissim, Maya Shloberg, Iris Biton, Yair Many, Adi Doron-Faigenboim, Hanita Zemach, Benjamin Avidan, Giora Ben-Ari.

**Formal analysis:** Giora Ben-Ari.

**Methodology:** Giora Ben-Ari.

**Project administration:** Yael Nissim, Giora Ben-Ari.

**Resources:** Ran Hovav, Giora Ben-Ari.

**Supervision:** Ran Hovav, Zohar Kerem, Giora Ben-Ari.

**Validation:** Yael Nissim, Giora Ben-Ari.

**Visualization:** Giora Ben-Ari.

**Writing – original draft:** Giora Ben-Ari.

**Writing – review & editing:** Giora Ben-Ari.

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
