## [Decision Letter · Decision Letter 0]

22 Oct 2019

PONE-D-19-22952

High temperature environment reduces olive oil yield and quality

PLOS ONE

Dear Dr. Ben-Ari,

Thank you for submitting your manuscript to PLOS ONE. After careful consideration, we feel that it has merit but does not fully meet PLOS ONE’s publication criteria as it currently stands. Therefore, we invite you to submit a revised version of the manuscript that addresses the points raised during the review process.

Please take care to make the edits suggested by the Reviewer below.

The use of supplementary information has helped to reduce the manuscript to a manageable size, however, as outlined by the Reviewer, the text is still too long in places: In the introduction and Discussion. There is no need to discuss the impacts of abiotic stress on species other than olives (this is well covered elsewhere). Don’t discuss any of your results in the introduction.

Avoid repeating results in the discussion section. Shorten the conclusions.

The description of the statistical analysis in the methods section is sufficient, but the explanations in the text are not always clear and should be revised. Similarly, in the tables/figures (including the supplementary figures), more explanation of the statistics is needed in the figure headings. The figure heading should be extensive enough that the figure can be read without referring to the text.

We would appreciate receiving your revised manuscript by Dec 06 2019 11:59PM. To enhance the reproducibility of your results, we recommend that if applicable you deposit your laboratory protocols in protocols.io, where a protocol can be assigned its own identifier (DOI) such that it can be cited independently in the future. For instructions see: http://journals.plos.org/plosone/s/submission-guidelines#loc-laboratory-protocols

We look forward to receiving your revised manuscript.

Kind regards,

Aidan D. Farrell, PhD

Academic Editor

PLOS ONE

**Journal Requirements:**

2. Thank you for stating that “The funders had no role in study design, data collection and analysis, decision to publish, or preparation of the manuscript” in your financial disclosure.

Please also provide the name of the funders of this study (as well as grant numbers if available) in your financial disclosure statement.

**Comments to the Author**

1. Is the manuscript technically sound, and do the data support the conclusions?

Reviewer #1: Yes

2. Has the statistical analysis been performed appropriately and rigorously? 

Reviewer #1: No

3. Have the authors made all data underlying the findings in their manuscript fully available?

Reviewer #1: No

4. Is the manuscript presented in an intelligible fashion and written in standard English?

Reviewer #1: Yes

5. Review Comments to the Author

Reviewer #1: Dear Authors

I have found the manuscript very interesting to gain deeper knowledge of the potential effects of increasing temperatures on olive yield and quality. An extensive experimental work has been carried out including a high number or evaluations under proper experimental design. However, I still consider there are some important flaws to be improved before acceptance:

The objectives of the work must be clearly presented in the last paragraph of the introduction section

Data analysis: were both 3 and 2-ways ANOVA carried out separately by years? Provide results of both years in Table S4. Not sure to understand the values of Table S4 b-e, could you explain?

The paper should be considerably shortened to be clearer and easier to follow:

- In my opinion the first part of the introduction is beyond the objectives of the work and could be substantially shorten

- There is no need to describe standard procedure in detail, you can only underline the main steps or variation from literature.

- Result section can be shortened by avoiding detailed description of figures. Only the most significant results should be described in text.

- Avoid reporting results again in the discussion section.

Minor comments:

Line 116: is 25 the most appropriate reference to describing these processes?

General results presented in lines 162-164 of Intro section should be deleted.

I wonder whether the Israel meteorological service stations (Lines 181-183) were located nearby the locations under study to be considered representative.

I strongly recommend you they use of quicker and easier technologies for measuring oil content in future studies, such as NMR or NIR methodologies.

Harvesting time and maturity index at this time are not clearly identified in the work. In L276 and L391 it is explained that “fruits were harvested at a maturity index of approximately 3”. However, a wide variability can be observed in Table S3, as much as from 1.12 (Coratina) to 3.71 (Barnea) in MT location in 2017. In my opinion, everything could had been easier by determining common dates for sampling and harvesting times in all cases regardless the maturity index, avoiding the need to follow maturity index, which, on the other hand, is currently considered not related to oil pattern accumulation. Please, consider this suggestion for future works.

Lines 431-434 Correct fruit weight values

Line 443: MT instead of HT

6. PLOS authors have the option to publish the peer review history of their article (what does this mean?). If published, this will include your full peer review and any attached files.

Reviewer #1: No

---

## [Author Response · Author response to Decision Letter 0]

18 Mar 2020

Dear Editor,

I'm sorry for the delay in sending the revised manuscript of our article "High temperature environment reduces olive oil yield and quality". I hope your comments and those of the reviewers have helped make the revised version worthy of publication.

Some paragraphs were deleted in response to comments. Additions made in the manuscript as well as responses to your and the reviewers comments, are highlighted in red font. 

Sincerely,

Giora Ben Ari

-PONE-D-19-22952

High temperature environment reduces olive oil yield and quality

PLOS ONE

The use of supplementary information has helped to reduce the manuscript to a manageable size, however, as outlined by the Reviewer, the text is still too long in places: In the introduction and Discussion. There is no need to discuss the impacts of abiotic stress on species other than olives (this is well covered elsewhere). Don’t discuss any of your results in the introduction.

Avoid repeating results in the discussion section. Shorten the conclusions.

The description of the statistical analysis in the methods section is sufficient, but the explanations in the text are not always clear and should be revised. Similarly, in the tables/figures (including the supplementary figures), more explanation of the statistics is needed in the figure headings. The figure heading should be extensive enough that the figure can be read without referring to the text.

Response: The entire manuscript has been reviewed and re-edited in accordance with your comments.

Reviewer #1: Dear Authors

I have found the manuscript very interesting to gain deeper knowledge of the potential effects of increasing temperatures on olive yield and quality. An extensive experimental work has been carried out including a high number or evaluations under proper experimental design. However, I still consider there are some important flaws to be improved before acceptance:

The objectives of the work must be clearly presented in the last paragraph of the introduction section

Response: Done.

Data analysis: were both 3 and 2-ways ANOVA carried out separately by years? Provide results of both years in Table S4. Not sure to understand the values of Table S4 b-e, could you explain?

Response: An explanation was added together with results for both years.

The paper should be considerably shortened to be clearer and easier to follow:

- In my opinion the first part of the introduction is beyond the objectives of the work and could be substantially shorten

Response: The first part of the introduction was substantially shortened. 

- There is no need to describe standard procedure in detail, you can only underline the main steps or variation from literature.

Response: The Materials and Methods section was substantially shortened by mentioning only the main steps and adding references to the relevant literature.

- Result section can be shortened by avoiding detailed description of figures. Only the most significant results should be described in text.

Response: Results section was shortened by avoiding detailed description of figures.

- Avoid reporting results again in the discussion section.

Response: Done.

Minor comments:

Line 116: is 25 the most appropriate reference to describing these processes?

Response: More relevant references were added.

General results presented in lines 162-164 of Intro section should be deleted.

Response: Deleted.

I wonder whether the Israel meteorological service stations (Lines 181-183) were located nearby the locations under study to be considered representative.

Response: Distances between meteorological stations and experiment locations were added.

I strongly recommend you they use of quicker and easier technologies for measuring oil content in future studies, such as NMR or NIR methodologies.

Response: We needed a methodology to quantify the exact amount of oil and the oil percentage of dry fruit in order to avoid the effect of fruit water content. Therefore, the suggested methods are not suitable for this study.

Harvesting time and maturity index at this time are not clearly identified in the work. 

In L276 and L391 it is explained that “fruits were harvested at a maturity index of approximately 3”. However, a wide variability can be observed in Table S3, as much as from 1.12 (Coratina) to 3.71 (Barnea) in MT location in 2017. In my opinion, everything could had been easier by determining common dates for sampling and harvesting times in all cases regardless the maturity index, avoiding the need to follow maturity index, which, on the other hand, is currently considered not related to oil pattern accumulation. Please, consider this suggestion for future works.

Response: This comment is correct. Unfortunately, my student harvested the Coratina olives in Tzuba too soon. Optimally, the olives at both sites should have been harvested at the same physiological stage.

Lines 431-434 Correct fruit weight values

Response: Corrected.

Line 443: MT instead of HT

Response: Corrected.

---

## [Editor Report · Decision Letter 1]

6 Apr 2020

High temperature environment reduces olive oil yield and quality

PONE-D-19-22952R1

Dear Dr. Ben-Ari,

We are pleased to inform you that your manuscript has been judged scientifically suitable for publication and will be formally accepted for publication once it complies with all outstanding technical requirements.

With kind regards,

Aidan D. Farrell, PhD

Academic Editor

PLOS ONE

Additional Editor Comments (optional):

Thank you for the revised manuscript. I am satisfied with the changes made and the manuscript now presents an acceptable summary of this extensive research.

Although the manuscript is still longer than normal, I appreciate the changes that have been made and am aware of the challenge of summarizing results from this many cultivars and parameters.

Some minor edits which you should attend to in the final stages:

Note the convention for reporting P -values, i.e. ‘Report exact p-values for all values greater than or equal to 0.001. P-values less than 0.001 may be expressed as p < 0.001, or as exponentials in studies of genetic associations.’

The image quality of the figures will need to be improved (as set out in the instructions to authors).
---

## [Editor Report · Acceptance letter]

10 Apr 2020

PONE-D-19-22952R1 

High temperature environment reduces olive oil yield and quality 

Dear Dr. Ben-Ari:

I am pleased to inform you that your manuscript has been deemed suitable for publication in PLOS ONE. Congratulations! Your manuscript is now with our production department. 

With kind regards,

on behalf of

Dr. Aidan D. Farrell 

Academic Editor

PLOS ONE